# Association between Parents’ Perceptions of Preschool Children’s Weight, Feeding Practices and Children’s Dietary Patterns: A Cross-Sectional Study in China

**DOI:** 10.3390/nu13113767

**Published:** 2021-10-25

**Authors:** Caihong Xiang, Youjie Zhang, Cuiting Yong, Yue Xi, Jiaqi Huo, Hanshuang Zou, Jiajing Liang, Zhiqian Jiang, Qian Lin

**Affiliations:** 1Department of Nutrition Science and Food Hygiene, Xiangya School of Public Health, Central South University, 110 Xiangya Road, Changsha 410078, China; xch0622@csu.edu.cn (C.X.); yongcuiting@csu.edu.cn (C.Y.); xiyue0404@csu.edu.cn (Y.X.); huojq0611@163.com (J.H.); zouhanshuang@csu.edu.cn (H.Z.); ljj1996@csu.edu.cn (J.L.); 2Department of Child and Adolescent Health and Social Medicine, School of Public Health, Medical College of Soochow University, 199 Ren Ai Road, Suzhou 215123, China; ujzhang@suda.edu.cn; 3Faculty of Agriculture, Life and Environment Science, Nutrition and Food Science Program, University of Alberta, 116st 85 Avenue, Edmonton, AB T6G 2R3, Canada; zhiqian3@ualberta.ca

**Keywords:** preschoolers, feeding practices, weight perceptions, dietary patterns

## Abstract

Parental perception of children’s weight may influence parents’ feeding practices, and in turn, child dietary intake and weight status; however, there is limited evidence generated for preschoolers. The aim of this cross-sectional study was to investigate associations between Chinese parents’ perceptions of child weight, feeding practices and preschoolers’ dietary patterns. Participants (1616 parent-child pairs) were recruited from six kindergartens in Hunan, China. Parents’ misperception, concern, and dissatisfaction on child weight were collected through a self-administered caregiver questionnaire. Parental feeding practices and children’s dietary intake were, respectively, assessed using the Child Feeding Questionnaire and a Food Frequency Questionnaire. Linear regression models were applied to analyze associations between parental weight perceptions, feeding practices, and preschooler’s dietary patterns. Associations between parents’ weight perceptions and dietary patterns were significant only among underweight children. Regardless of child weight status, parental weight underestimation and preference for a heavier child were positively associated with pressure-to-eat. Parental weight concern was positively associated with restriction in normal weight child, but this was not found in other weight groups. In conclusion, Parents’ misperception, concern, and dissatisfaction about child weight are associated with parents’ feeding practices and may influence preschoolers’ dietary quality, but the relationships vary by children’s actual weight status.

## 1. Introduction

The prevalence of obesity in preschool children has increased sharply, and it has become a major global public health issue [1]. Data from the Chinese Center for Disease Control and Prevention shows that the obesity rate in Chinese children under 7 years old has increased from 0.9% to 8.4% from 1985 to 2015 [2,3]. Childhood obesity not only increases the risk of obesity later in life [4], it also increases the risk of chronic diseases such as cardiovascular disease and type 2 diabetes [5,6].

Imbalanced diet, such as the high-sugar and high-fat diet, has been identified as an important cause of obesity [7,8]. Since the availability of food products has improved in China, children’s eating behavior has changed dramatically. Daily eating occasions and energy intake from snacks has increased significantly among Chinese preschoolers [9]. The dietary pattern characterized by sugary drinks, savory snacks, and fast foods has become prevalent among Chinese preschoolers, which is associated with a higher risk of adiposity as compared to the traditional dietary pattern [10].

Parents play an important role in shaping young children’s dietary intake. Their feeding attitudes and practices influence children’s food intake and weight status [11]. Findings based on the widely applied Child Feeding Questionnaire (CFQ) indicate that feeding practices at high levels of control can compromise children’s capability to self-regulate food intake and cause problematic eating behaviors such as picky eating [12,13,14,15,16]. Factors associated with parental feeding practices would provide important implications for advocating desirable feeding practices that help with establishing health eating behaviors among the young.

Parents’ feeding practices have shown to be associated with various types of parental perceptions of children’s weight status. Parents who perceived their children as overweight/obese are more likely to monitor and restrict children’s food intake [17,18,19], while parents who perceived their children as underweight are more likely to pressure their children to eat more [17,20]. Parental concern and dissatisfaction regarding children’s weight status have also found to be associated with the feeding practices of restriction and pressure to eat [21,22]. Despite abundant studies reporting the relationships between parental weight perception and feeding practices, only a few of them has considered its interplay with the accuracy of weight perception. However, about 50–90% of parents misclassify children’s weight status, especially for young children [23,24,25]. Parents’ misperception of children’s weight status reflects their dissatisfaction and concern [21,26], and may influence parental feeding practices [27], children’s dietary quality [28], and weight status [29,30]. Interactions among parental perceptions of child weight, feeding practices, and child dietary intake deserve further investigation.

Chinese parents are often characterized by high levels of control, and the prevalence of parental misperception of child body weight is much higher in China than other countries [23]. Existing studies have investigated associations between parents’ child weight perceptions and their parenting practices among school-aged children and adolescents [31,32,33]. Only a few are conducted among Chinese preschoolers. A study of 176 Chinese parent-preschooler pairs shows that maternal perceptions of child actual and ideal body shape, and child dietary intake vary across clusters of parental feeding practices [34]. However, the relationships between parental weight perceptions and feeding practices are not fully explored.

Comparing to school-aged children, preschoolers’ dietary intake is under greater parental influence and their eating behavior is more malleable. However, less is known regarding the influence of parental weight perceptions and feeding practices on preschoolers’ dietary quality. Therefore, the current study aims to examine the inter-relationships between parental child weight perceptions (misperception, concern, and dissatisfaction), feeding practices, and children’s dietary intake among Chinese preschoolers and their parents. It can help understand the factors in the family environment that influence the formation of preschoolers’ diet patterns, and further provide guidance and recommendations on the management of childhood obesity.

## 2. Materials and Methods

### 2.1. Study Design and Sample

In this cross-sectional study, we used random cluster sampling to select 6 public kindergartens in Changsha City, Hunan Province from January to March 2021. Parent-child pairs in kindergartens were screened using the following criteria: the participating parents (1) were responsible for their children’s diet for more than six months, (2) could read and fully understand the questionnaire, and (3) the children were between the ages of 2 and 7. Directors from participating kindergartens helped by distributing electronic questionnaires to eligible parents using an online survey tool (www.wjx.cn accessed on 4 March 2021). After reading and signing the electronic consent form, parents completed the self-administrated questionnaires. The study was approved by the Ethics Review Committee of the Xiangya School of Public Health, Central South University (No. XYGW-2020-105).

### 2.2. Variables and Measurement

#### 2.2.1. Demographics and Anthropometrics

We collected family demographic information using self-report questions, including child age, gender, number of siblings, and parental role (mother or father), age, and education level.

We estimated child BMI based on parent-reported height and weight. According to the WHO growth reference for children and adolescents, Child weight status was classified as obese (BMI ≥ +2 SD), overweight (BMI ≥ +1 SD), normal weight (BMI > −1 SD to <+1 SD), and underweight (BMI ≤ −1 SD) [35].

#### 2.2.2. Parental Perceptions of Child Weight

Concern about child becoming overweight. Parental concern about children weight was measured using an item from the CFQ: “how concerned are you about your child becoming overweight?” with response options of “unconcerned”, “a little concerned”, “concerned”, “fairly concerned”, and “very concerned” [12]. The response options were then dichotomized into to the unconcerned (“unconcerned”) and concerned (the remaining) groups.

Misperception of child body weight. Parental perception of child current weight status was also assessed using an item from the CFQ: “How would you describe your child’s weight currently?”, with response options of “markedly underweight”, “underweight”, “normal weight”, “overweight”, and “markedly overweight”. The responses options were recorded into underweight (“very underweight” and “underweight”), normal weight, overweight (“very overweight” and “overweight”) groups. Compared with child weight status generated from estimated BMI, paternal misperception of child body weight was categorized as underestimation, overestimation, and correct estimation.

Satisfaction with child body weight. Caregivers were shown a chart with seven silhouettes sorted from lightest to heaviest body weight according to their child’s gender and asked to choose the one that best matched their child’s current figure and the one that would be the ideal figure for the child. The body silhouettes were from our previous study by Tang et al., aimed to assess parental visual perception of their child’s weight [36]. The difference between ideal figure and current figure indicates parents’ satisfaction with children’s body weight [37,38,39]. The difference could vary between −6 and 6, with a negative difference indicating that parents wish their child to be thinner, a positive difference indicating that parents wish their child to be heavier, and 0 indicating that parents wish their child to stay the same.

#### 2.2.3. Parental Feeding Practices

Parental feeding practices were assessed using four subscales from the Chinese version CFQ, including restriction (four questions, e.g., “I have to be sure that my child does not eat too many high-fat foods.”), food as rewards (two questions, e.g., “I offer my child her favorite foods in exchange for good behavior.”), pressure to eat (four questions, e.g., “My child should always eat all of the food on her plate.”) and monitoring (four questions, e.g., “How much do you keep track of the high-fat foods that your child eats?”) [40]. The responses to each question are based on a 5-point Likert scale expressing agreement (1 = disagree to 5 = agree) or frequency (1 = never to 5 = always), depending on the subscale. The mean score of the items that comprise each subscale is the factor score, with higher scores indicating more controlling feeding practices. Cronbach’s α in the present study were 0.814, 0.789, 0.671, and 0.916, respectively.

#### 2.2.4. Child Dietary Pattern

Parents reported their children’s dietary intake using a 27-item semi-quantitative food frequency questionnaire (FFQ) [41]. The FFQ assessed food intake frequency in the past 3 months with response options of “never or less than once/month, 1–3 times/month, 1–2 times/week, 3–4 times/week, 5–6 times a week, 1 time/day, 2 times/day, and 3 times/day or more”, then converted to 0, 0.5, 1.5, 3.5, 5.5, 7, 14, and 21 times per week, respectively.

By combining items of the same category of food in the food frequency table, 20 food categories were finally obtained. Dietary patterns were identified by factor analysis using the principal component method with varimax rotation. Factor analysis is a classic method to identify dietary patterns, and is widely used in Chinese preschool children [10,42,43,44]. Kaiser-Meyer-Olkin (KMO) = 0.926, and Bartlett’s sphericity test *p* < 0.001, indicating that the sample was suitable for factor analysis. Based on eigenvalues (>1) and the scree plot, a two-factor structure was extracted with 53.1% of the cumulative variance explained. Table 1 shows factor loadings of each food group. A factor loading >0.4 indicated a strong association with the factor. According to the characteristics of food groups, the two dietary patterns were identified as traditional dietary pattern (factor 1, contribution rate of 35.6%) and snacking dietary pattern (factor 2, contribution rate of 17.5%). After the food frequency was converted to the standard normal distribution and weighted by factor loading, the standardized scores of the two dietary patterns of each survey object were obtained.

### 2.3. Statistical Analysis

The data analysis was conducted using IBM SPSS26.0 software (IBM Corp., Armonk, NY, USA), and the figure was created using GraphPad Prism8.3.0 software (La Jolla, CA, USA). Categorical variables were presented in numbers and percentages, and continuous variables were presented in means with standard deviations (SD). Models examining associations between parental weight perceptions and child dietary patterns included parental concern about child becoming overweight, satisfaction with child body weight, and misperception of child body weight as independent variables, and preschoolers’ traditional dietary pattern standardized score and snacking dietary pattern standardized score as dependent variables, respectively. Models examining associations between parental weight perceptions and parental feeding practices included parental concern about child becoming overweight, satisfaction with child body weight, and misperception of child body weight as independent variables and parental scores of restriction, food as reward, pressure-to-eat, and monitoring as dependent variables, respectively. Models examined associations between parental feeding practices and child dietary patterns included parental scores of restriction, food as reward, pressure-to-eat, and monitoring as independent variables and preschoolers’ traditional dietary pattern standardized score and snacking dietary pattern standardized score as dependent variables, respectively. All the models were stratified by child weight status and adjusted for child gender, age, and only child, and parental role, and education. Regression coefficients (β) and 95% confidence intervals (95% CI) were identified. A two-tailed *p* < 0.05 was considered statistically significant in all analyses. Participants who did not provide their height and weight (*n* = 128, 7.9% of total sample) were excluded from the final analysis.

## 3. Results

### 3.1. Sample Characteristics

Sample characteristics are shown in Table 2. A total of 1616 parent-child pairs participated in the study. For children, the mean (SD) age was 4.54 ± 0.85 years and 53.7% were boys. The prevalence of overweight and obesity was 16.3% and 7.6%, respectively. For parents, the majority (88.8%) were mothers. The mean (SD) age of mothers was 34.6 ± 4.3 years, and 62.9% of the mothers had college education or above. The mean (SD) age of fathers was 37.2 ± 5.4 years, and 64.7% of them had college education or above.

### 3.2. Parents’ Weight Perceptions

Only 60.5% (*n* = 900) of parents correctly assessed their children’s body weight, and 29.4% (*n* = 437) underestimated their children’s body weight. Misperception of body weight was related with the children’s age, gender, and actual weight status. Parents of younger children, boys, or children with overweight and obesity were significantly more likely to underestimate child weight status (Table 2).

Parental weight perceptions by children’s actual weight status are shown in Figure 1. The heavier the child in regards to their weight status, the higher the proportions of parents concerning about child becoming overweight (F = 38.82, *p* < 0.001), wishing child to become thinner (F = 174.93, *p* < 0.001), and underestimating child body weight (F = 1220.09, *p* < 0.001).

### 3.3. Parental Weight Perceptions and Childrens’ Dietary Patterns

Associations between parental weight perceptions and child dietary patterns were found only in children with underweight (Table 3, Appendix A). Concerning about child becoming overweight and wishing a thinner child were positively associated with preschoolers’ snacking eating pattern scores (*p* = 0.007, 0.029, respectively). Wishing a heavier child was inversely associated with preschoolers’ traditional eating pattern scores (*p* = 0.009). Overestimating child weight status was positively associated with preschoolers’ traditional eating pattern scores (*p* = 0.020).

Regression model with adjustments of child gender, age, only child, parental role, and parental education.

### 3.4. Parents’ Weight Perceptions and Feeding Practices

Table 4 and Appendix A show the associations between parental perceptions of child weight and feeding practices by child weight status. In all child weight status groups, parents who wished their child to be heavier reported higher levels of pressure to eat (*p* = 0.001, 0.002, 0.006, respectively). After adjusting for child gender, age, only child, caregiver and parental education, these associations remain significant (*p* = 0.001, 0.001, 0.004, respectively).

For children with underweight, parents concerned about child becoming overweight reported lower levels of restriction and monitoring (*p* = 0.046, 0.029, respectively).

For children with normal weight, parents concerned about the child becoming overweight reported higher levels of restriction (*p* = 0.033), and parents who underestimated child’s weight reported higher level of pressure-to-eat (*p* = 0.017).

For children with overweight/obesity, parents who underestimated child’s weight reported lower level of restriction and higher level of pressure-to-eat (*p* = 0.046, 0.002, respectively).

### 3.5. Caregiver Feeding Practices and Children’S Dietary Patterns

Table 5 and Appendix A show relationships between parental feeding practices and children’s dietary patterns. Restriction was inversely associated with the snacking dietary pattern across child weight groups (*p* = 0.012, 0.017, and <0.001, respectively), but was not associated with traditional dietary pattern. Food-as-reward was only inversely associated with traditional dietary pattern in children with normal weight (*p* = 0.038). Pressure-to-eat was positively associated with snacking dietary pattern in children with normal weight (*p* = 0.008) and was inversely associated with traditional dietary pattern in children with underweight (*p* = 0.029). Monitoring was inversely associated with snacking dietary pattern in children with underweight and normal weight (*p* = 0.006, <0.001, respectively) and was positively associated with traditional dietary pattern across child weight groups (*p* = 0.012, 0.008, and 0.017, respectively).

## 4. Discussion

This study sought to examine whether and how parental perceptions about child’s weight are related to feeding practices and dietary patterns in preschool children. Our results suggest that parental weight perceptions are directly associated with dietary patterns only in children with underweight, but parental perceptions could affect feeding practices, and in turn, children’s diet.

In this study, parental weight perceptions (misperception, satisfaction, and concern) were all correlated with the children’s actual body shape. Parents were more likely to worry about overweight/obese children gaining weight and want obese/overweight children to lose weight. This suggests that parents were generally aware of the consequences of overweight/obesity in children and are motivated to take weight-control actions. However, most parents (72.1%) of overweight and obese children underestimated their child weight status. It is close to the prevalence (69%) found among parents of 6–18 years old children with overweight/obesity from the 2006 China Health and Nutrition Survey [45], and both are higher than the prevalence reported in a meta-analysis of international studies (50.7%) [23]. The visual normalization theory suggests that the high prevalence of overweight/obesity in children is an important reason for parental underestimation of plus body shapes in children, which in turn impedes the management of childhood obesity [46]. Consistent with other studies [33,47,48], we found that the age and gender of children were important factors affecting parental misperception of children’s weight. Younger children or boys’ weights were more likely to be underestimated, whereas girls’ weights were more likely to be overestimated by their parents. This reflects the society’s preferences towards chubby kids, bulky boys, and slender girls [33,46,48]. Our findings suggest a need to guide the public with correct weight perceptions and healthy weight preferences.

Weight overestimation was positively associated with traditional eating patterns among underweight children. The traditional dietary pattern is relatively healthier than the snacking dietary pattern. A previous study found that regardless of the actual weight, Chinese parents were more likely to perceive their children’s weight as healthy as long as their children behaved healthily [33]. This may help explain the positive association between weight overestimation and traditional dietary pattern among children with underweight. However, this phenomenon was not found among children with overweight/obesity. In general, we observed only a few associations of parental child weight perception with preschoolers’ dietary patterns, but more with parental feeding practices.

Weight underestimation was positively associated with parental pressure-to-eat. Not surprisingly, parents who thought their children being underweight were more likely to pressure them to eat more [17,20]. Pressure-to-eat was generally considered a problematic feeding behavior that could lead children to make more negative comments about pressured foods (usually healthy foods, such as vegetables and fruits), avoid consuming these foods, and develop problematic eating behaviors such as picky and partial eating [13,14,49]. In our study, normal weight children who were pressured to eat had significantly higher scores for snacking eating patterns, which can increase the risk of unhealthy weight gain [50].

In addition, we found that parents who underestimated their overweight/obese child’s body size were less likely to restrict their diet. This finding is concordant with Yilmaz et al.’s study of preschoolers aged 5 to 7, which reported that when mother’s perception of her child’s nutritional status is incorrectly low, they showed less restriction and make the child eat freely [27]. Similarly, the present study found that parental restrictive feeding was inversely associated with preschoolers’ snacking dietary pattern. However, longitudinal and lab-based studies tend to believe that children under greater parental restrictions are more likely to eat unhealthy foods [51]. This prospective relationship needs additional considerations on its interactions with child actual weight and parental weight misperceptions.

We found that parental concern about child becoming overweight was positively associated with snacking dietary pattern scores among preschoolers with underweight. Other studies found that parental concern for child becoming overweight was positively associated with sugar-sweetened beverages intake and was significantly predicted by parental concerns about child’s dietary quality [52,53]. Parents may be concerned about their children’s weight because of their high intake of snacks in present study. Interestingly, parents who concerned about their children becoming overweight reported lower levels of parental restrictive and monitoring feeding behaviors for preschoolers with underweight, which was opposite to the relationship found for normal weight preschoolers and finding from previous studies [20,38,54,55,56]. One possible explanation is that even if parents are concerned about children becoming overweight, they may still want their underweight children to gain more weight so that they may employ less restriction and monitoring. In addition, other factors such as child temperament, may play a role, but are not included in the present study.

In the present study, parents wishing their children to be heavier were more likely to use the feeding practice of pressure to eat, regardless of the children’s weight status. Consistent results were found in a recent birth cohort study (*n* = 3233), in which mothers who wished their children to be heavier used more frequent pressuring feeding at both ages 4 and 7 [21]. These findings suggest that parental desire for a heavier child motivates inappropriate controlling feeding practices which have shown to be associated with less healthy food intake and increased obesity risk [50,57]. Parents’ dissatisfaction with their child’s body weight could stem from dissatisfaction with their own weight [58], which have been listed as diagnostic features in a number of psychopathologies, was also found to be associated with parents’ feeding, children’s dietary intake and weight [39]. Interventions aimed at parents to establish healthy body image attitudes and feeding practices might be beneficial for childhood obesity management.

We found that parental inappropriate feeding practices, include pressure-to-eat and food-as-forward were associated with increased risks for children’s unhealthy dietary patterns, whereas parental restriction of feeding to be associated with greater intake of healthy foods. Pressure and restriction were both referred to direct controlling feeding practices, and have been shown to undermine the child’s internal control over food intake as well as child’s mental health, including negative self-evaluation [59,60]. Although our study found that children with greater parental restriction were less likely to developing a snacking dietary pattern, they may have unhealthy food intake rebound without parental restriction. Thus, we hesitate to suggest parents to employ the strategy of restriction that is known to be generally deleterious. Food as reward is another type of unhealthy feeding separated from restriction dimension in the original CFQ. Parents who rewarded child with snacks could temporarily exchange heathy food for their child, but over time this inappropriate feeding strategy increased food aversion and lead to poor eating habits [15]. In Levene and Williams’ feeding strategies for parents/caregivers, food as reward should be avoided [61]. Monitoring was referred to indirect control over unhealthy food for children, often linked to children consuming fewer sweets [62,63]. Parental monitoring feeding may be a better method for improving a child’s diet than attempts at restriction.

The current study has several limitations. First, we used a cross-sectional design to investigate the relationship between parental perceptions of children’s weight, feeding practices, and children’s dietary patterns, and it was unable to demonstrate cause-effect relationship. Future studies will need to use longitudinal designs to further verify the relationship. Second, the children’s height and weight were self-reported by parents which could introduce recall bias. In addition, the parent-child pairs in this study were from a single city in central south China, which does not have national representativeness.

## 5. Conclusions

With a sufficient sample size, standardized assessment tools, and appropriate statistical methods, our study demonstrated parental perceptions of child’s weight in relation to both parental feeding practices and preschooler’s dietary patterns, which has been investigated only in very limited studies. Despite that significant associations between parental weight perceptions and child dietary patterns only existed among preschoolers with underweight, parental weight perceptions of underestimating child weight status and wishing child to be heavier showed robust associations with pressure-to-eat among children in all weight groups, and pressure-to-eat demonstrated unfavorable associations with preschoolers’ dietary patterns. These incorrect parental perceptions about child weight may adversely influence children’s dietary intake through motivating undesirable feeding practices. Future studies need to apply longitudinal and experimental designs to confirm the potential effect of parental child weight perceptions and its application on healthy eating and healthy weight promotion among Chinese preschoolers and their parents.

## Figures and Tables

**Figure 1 nutrients-13-03767-f001:**
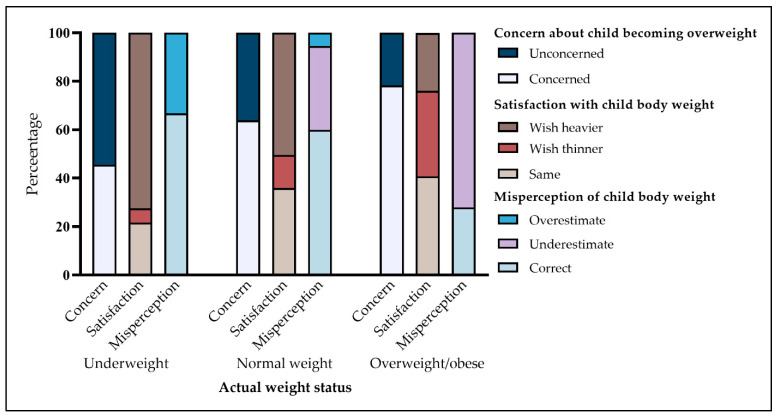
Parental weight perceptions and actual weight status of preschool children.

**Table 1 nutrients-13-03767-t001:** Factor loadings for 2 dietary patterns.

Food Group	Traditional Dietary Pattern	Snacking Dietary Pattern
Grains	**0.565**	−0.044
Roots and tubers	**0.607**	0.213
Vitamin A–rich fruits	**0.702**	0.078
other fruits	**0.715**	0.051
Vitamin A–rich vegetables	**0.783**	−0.038
Other vegetables	**0.798**	0.024
Legumes	**0.777**	0.127
Nuts	**0.637**	0.272
Eggs	**0.670**	0.036
Livestock meat	**0.685**	0.122
Poultry meat	**0.640**	0.189
Processed meat	**0.477**	0.523
Fish or shellfish	**0.566**	**0.430**
Dairy products	**0.470**	0.246
Dessert	0.176	**0.716**
Puffed food	0.032	**0.830**
Spicy snacks	0.068	**0.844**
Sugar-sweetened beverages	0.041	**0.824**
Fresh fruit and vegetable juices	0.177	**0.662**
Fast food	−0.007	**0.881**

Bolding indicates factor loading > 0.4.

**Table 2 nutrients-13-03767-t002:** Characteristics of children and parents (*n* = 1616, *n*(%) or Mean(SD)).

Characteristics	Total	Misperception of Child Body Weight *			*p*
		Underestimate *n* = 437	Correct Perception *n* = 900	Overestimate *n* = 151	
**Child characteristics**					
Age, mean (SD)	4.54 (0.85)	4.51 (0.83)	4.54 (0.86)	4.76 (0.91)	**0.005**
Gender					**0.030**
Boy	867 (53.7)	153 (31.6)	457 (57.4)	88 (11.0)	
Girl	749 (46.3)	184 (26.7)	441 (64.1)	63 (9.2)	
Only child					0.187
Yes	623 (38.6)	277 (30.6)	530 (58.7)	97 (10.7)	
No	993 (61.4)	160 (27.4)	370 (63.4)	54 (9.2)	
Weight status *					**<0.001**
Underweight	222 (14.9)	0 (0.0)	93 (41.9)	129 (58.1)	
Normal weight	911 (61.2)	148 (16.2)	741 (81.3)	22 (2.5)	
Overweight	242 (16.3)	208 (86.0)	34 (14.0)	0 (0.0)	
Obesity	113 (7.6)	81 (71.7)	32 (28.3)	0 (0.0)	
**Caregiver characteristics**					
Caregiver					0.721
Mother	1435 (88.8)	385 (29.1)	806 (60.8)	134 (10.1)	
Father	181 (11.2)	53 (31.9)	94 (57.7)	17 (10.4)	
Maternal age, mean (SD)	34.62 (4.28)	34.57 (4.34)	34.69 (4.25)	34.31 (4.51)	0.598
Maternal education					0.390
Junior high school or below	53 (3.3)	8 (20.0)	28 (70.0)	4 (10.0)	
High school	186 (11.5)	59 (34.3)	95 (55.2)	18 (10.5)	
College or above	1377 (85.2)	370 (29.0)	777 (60.9)	129 (10.1)	
Paternal age, mean (SD)	37.17 (5.35)	37.31 (5.23)	37.10 (5.36)	37.05 (5.84)	0.772
Paternal education					0.341
Junior high school or below	62 (3.8)	14 (25.9)	31 (57.4)	9 (16.7)	
High school	182 (11.3)	54 (33.8)	93 (58.1)	13 (8.1)	
College or above	1372 (84.9)	369 (29.0)	776 (60.9)	129 (10.1)	
**Parental feeding practices**					
Restriction, mean (SD)	3.92 (0.83)	3.88 (0.83)	3.97 (0.82)	3.83 (0.90)	0.061
Food as rewards, mean (SD)	3.71 (0.94)	3.79 (0.88)	3.73 (0.95)	3.54 (0.95)	**0.019**
Pressure to eat, mean (SD)	3.06 (0.80)	3.10 (0.79)	3.03 (0.82)	3.06 (0.81)	0.404
Monitoring, mean (SD)	3.85 (0.83)	3.82 (0.84)	3.87 (0.83)	3.84 (0.81)	0.599
**Child dietary pattern scores**					
traditional dietary pattern, mean (SD)	0.00 (1.00)	−0.06 (0.80)	0.03 (1.09)	0.19 (1.13)	**0.032**
snacking dietary pattern, mean (SD)	0.00 (1.00)	0.02 (0.95)	−0.35 (0.97)	0.02 (0.74)	0.535

* 128 missing data. Bolding indicates statistically significant values, *p* < 0.05.

**Table 3 nutrients-13-03767-t003:** Linear regression relationships between parental weight perceptions and children’s dietary patterns among different ZBMI weight status of children.

	Traditional Dietary Pattern β (95%CI)	Snacking Dietary Pattern β (95%CI)
**Underweight**		
**Concern about child becoming overweight**		
Unconcerned	1	1
Concerned	−0.092 (−0.369, 0.185)	**0.276 (0.075, 0.477) ****
**Satisfaction with child body weight**		
Same	1	1
Wish thinner	**−0.627 (−1.242, −0.013) ***	**0.506 (0.052, 0.960) ***
Wish heavier	**−0.428 (−0.749, −0.106) ****	−0.039 (−0.277, 0.199)
**Misperception of child body weight**		
Correct	1	1
Overestimate	**0.319 (0.051, 0.588) ***	0.061 (−0.139, 0.262)
**Normal weight**		
**Concern about child becoming overweight**		
Unconcerned	1	1
concerned	0.092 (−0.055, 0.239)	0.031 (−0.104, 0.166)
**Satisfaction with child body weight**		
Same	1	1
Wish thinner	−0.076 (−0.300, 0.149)	0.115 (−0.091, 0.321)
Wish heavier	−0.139 (−0.293, 0.016)	0.080 (−0.061, 0.221)
**Misperception of child body weight**		
Correct	1	
Overestimate	0.244 (−0.215, 0.702)	0.169 (−0.252, 0.589)
Underestimate	−0.149 (−0.340, 0.042)	−0.078 (−0.253, 0.098)
**Overweight/Obese**		
**Concern about child becoming overweight**		
Unconcerned	1	1
Concerned	−0.112 (−0.322, 0.098)	0.181 (−0.058, 0.420)
**Satisfaction with child body weight**		
Same	1	1
Wish thinner	−0.115 (−0.316, 0.086)	0.007 (−0.222, 0.235)
Wish heavier	0.003 (−0.220, 0.226)	−0.172 (−0.426, 0.082)
**Misperception of child body weight**		
Correct	1	1
Underestimate	0.004 (−0.221, 0.229)	0.176 (−0.079, 0.432)

Bolding indicates statistically significant values, * *p* < 0.05, ** *p* < 0.01. Regression model with adjustments of child gender, age, only child, parental role and parental education.

**Table 4 nutrients-13-03767-t004:** Linear regression relationships between parental weight perceptions and caregiver feeding practices among different ZBMI weight status of children.

Weight Perceptions	Restriction	Food as Reward	Pressure to Eat	Monitoring
	β (95%CI)	β (95%CI)	β (95%CI)	β (95%CI)
**Underweight (*n* = 222)**				
**Concern about child becoming overweight**				
Unconcerned	1	1	1	1
Concerned	**−0.244 (−0.484, −0.005) ***	−0.154 (−0.418, 0.111)	−0.119 (−0.337, 0.098)	**−0.247 (−0.468, −0.026) ***
**Satisfaction with child body weight**				
Same	1	1	1	1
Wish thinner	0.109 (−0.433, 0.651)	−0.051 (−0.646, 0.543)	−0.097 (−0.572, 0.378)	0.057 (−0.445, 0.559)
Wish heavier	0.236 (−0.048, 0.520)	0.259 (−0.052, 0.571)	**0.443 (0.192, 0.692) ****	0.214 (−0.049, 0.477)
**Misperception of child body weight**				
Correct	1	1	1	1
Overestimate	−0.180 (−0.417, 0.056)	−0.171 (−0.430, 0.089)	−0.204 (−0.417, 0.008)	−0.127 (−0.346, 0.092)
**Normal weight (*n* = 911)**				
**Concern about child becoming overweight**				
Unconcerned	1	1	1	1
Concerned	**0.123 (0.010, 0.236) ***	0.002 (−0.124, 0.129)	0.025 (−0.084, 0.135)	0.030 (−0.082, 0.143)
**Satisfaction with child body weight**				
Same	1	1	1	1
Wish thinner	−0.007 (−0.180, 0.166)	−0.023 (−0.216, 0.170)	−0.102 (−0.268, 0.064)	0.015 (−0.157, 0.186)
Wish heavier	−0.101 (−0.219, 0.018)	0.022 (−0.111, 0.154)	**0.187 (0.073, 0.301) ****	−0.010 (−0.128, 0.107)
**Misperception of child body weight**				
Correct	1	1	1	1
Overestimate	0.290 (−0.062, 0.643)	0.258 (−0.136, 0.652)	0.109 (−0.232, 0.450)	0.288 (−0.062, 0.638)
Underestimate	−0.120 (−0.267, 0.027)	0.057 (−0.107, 0.221)	**0.174 (0.032, 0.316) ***	0.042 (−0.104, 0.188)
**Overweight/Obese (*n* = 355)**				
**Concern about child becoming overweight**				
Unconcerned	1	1	1	1
Concerned	0.052 (−0.153, 0.258)	−0.141 (−0.347, 0.091)	−0.146 (−0.350, 0.057)	−0.034 (−0.253, 0.186)
**Satisfaction with child body weight**				
Same	1	1	1	1
Wish thinner	0.168 (−0.027, 0.363)	−0.013 (−0.236, 0.210)	−0.032 (−0.224, 0.160)	0.162 (−0.048, 0.371)
Wish heavier	0.077 (−0.294, 0.140)	−0.062 (−0.310, 0.186)	**0.319 (0.106, 0.533) ****	0.140 (−0.093, 0.373)
**Misperception of child body weight**				
Correct	1	1	1	1
Underestimate	**−0.223 (−0.441, −0.004) ***	0.059 (−0.190, 0.309)	**0.345 (0.130, 0.560) ****	−0.102 (−0.377, 0.132)

Bolding indicates statistically significant values, * *p* < 0.05, ** *p* < 0.01. Regression model with adjustments of child gender, age, only child, parental role and parental education.

**Table 5 nutrients-13-03767-t005:** Linear regression relationships between parental feeding practices and children’s dietary patterns among different ZBMI weight status of children.

	Traditional Dietary Pattern β (95%CI)	Snacking Dietary Pattern β (95%CI)
**Underweight**		
Restriction	0.051 (−0.124, 0.226)	**−0.160 (−0.285, −0.035) ***
Food as reward	−0.001 (−0.143, −0.141)	−0.025 (−0.127, 0.076)
Pressure to eat	**−0.191 (−0.362, −0.020) ***	0.035 (−0.087, 0.157)
Monitoring	**0.241 (0.054, 0.429) ***	**−0.189 (−0.322, −0.056) ****
**Normal weight**		
Restriction	−0.001 (−0.098, 0.100)	**−0.109 (−0.198, −0.019) ***
Food as reward	**−0.084 (−0.163, −0.005) ***	−0.043 (−0.114, 0.028)
Pressure to eat	0.012 (−0.077, 0.102)	**0.110 (0.029, 0.191) ****
Monitoring	**0.135 (0.035, 0.236) ****	**−0.172 (−0.263, −0.082) ****
**Overweight/Obese**		
Restriction	0.010 (−0.122, 0.142)	**−0.259 (−0.403, −0.114) ****
Food as reward	−0.059 (−0.159, 0.041)	−0.074 (−0.184, 0.036)
Pressure to eat	0.015 (−0.096, 0.126)	0.070 (−0.051, 0.191)
Monitoring	**0.148 (0.027, 0.269) ***	−0.116 (−0.248, 0.016)

Bolding indicates statistically significant values, * *p* < 0.05, ** *p* < 0.01. Regression model includes restriction, food as reward, pressure to eat and monitoring, plus child gender, age, only child, parental role and parental education.

## Data Availability

The data that support the findings of this study are not publicly available due to the data containing information that could compromise participant privacy but are available from the corresponding author on reasonable request.

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
