# Peer review of "Association between Parents’ Perceptions of Preschool Children’s Weight, Feeding Practices and Children’s Dietary Patterns: A Cross-Sectional Study in China"

_nutrients, 2021, doi:10.3390/nu13113767_

Round 1
Reviewer 1 Report
This study aimed at investigating the association between parental perceptions of their child’s weight, parental feeding practices and children’s dietary patterns. The article explores an interesting and important topic and has several strengths.I have listed my questions and comments:
- The abstract starts directly with results without mentioning the analysis done in the methods section. I suggest adding a sentence with the analytical approach and methods used to obtain the results presented.
- Are the regression coefficients standardized? I suggest to either specifying that they are indeed standardized (if it is the case), or avoid using the symbol for non-standardized coefficients. In addition, if these coefficients are not standardized, I suggest standardizing them so the magnitude of the prediction is clearer.
- Why did the authors use a variable center approach (factor analysis) to identify dietary patterns instead of a person center approach (latent class or profile analyses) that seems more appropriate to uncover individual behavioral patterns? If the factor analysis will be kept instead of an LCA or LPA, I suggest including a brief justification for selecting this method.
- Even though it is a cross-sectional study, I recommend including more detail of the analyses done in terms of dependent and independent variables for each model, as the statistical analysis section should be clear enough instead of relying on the results tables’ structure.
- The authors reported that 128 participants had missing data in BMI. Were the rest of the variables and covariates complete or they had missing data as well? If the latter, please specify how were missing data addressed. Additionally, since missing data were dealt with deletion instead of with imputation, please justify this decision.
- I suggest using person-first language where the person is first rather than identity-first (e.g., use “children with obesity” instead of “obese children”).
- Finally, proofread the paper to address some minor issues and improve readability (e.g., delete “who” in lines 214 and 216).
Reviewer 2 Report
This manuscript presents the findings of a cross-sectional study conducted in a sample of 1616 parent-child pairs in China and examined the associations between parents’ perceptions of child weight (misperception, concern, and dissatisfaction), feeding practices and preschoolers’ dietary patterns. The study methods are well described, and the results highlight that incorrect parental perceptions about child weight may lead to undesirable feeding practices and thus adversely influence children’s dietary intake.
The topic is highly interesting, the manuscript is well-written
I only have a comment on the derived dietary patterns. Could the authors please confirm that the “Processed meat” is part of the traditional instead of the snacking dietary pattern? It seems that the factor loading is higher in the 2nd pattern (0.477 vs 0.523), so I would assume that since this food group had a factor loading value >0.4 in both components, it would be used to name the component for which it had the higher factor loading value.
